# Effect of rising fuel prices on small-scale fisheries livelihoods and marine sustainability in Ghana

**Victor Owusu** *

Department of Geography Education, University of Education, Winneba, Ghana

* vowusu@uew.edu.gh

## Abstract

This study investigates the effects of fuel price hikes on the livelihoods of small-scale coastal fisherfolk in Ghana. The study applied a mixed-methods approach consisting of a questionnaire survey of 320 fisherfolk and 20 interviews with stakeholders in the fisheries sector. Increase in expenses, reduced frequency of fishing, an upsurge in social vices, and declining small-scale fisheries opportunities were found to be the main effects of fuel price hikes on fisherfolk livelihoods. The results reveal that fisherfolk experienced financial, emotional and psychological shocks due to the high cost of fuel. Dependency on savings, borrowing, petty trading, migration and farming were found to the main coping strategies. However, the various livelihood coping strategies deployed by fisherfolk were not sufficient to ameliorate their economic hardship. The findings show that fuel price hikes can contribute to reduction in fishing pressure and overcapacity despite the current socioeconomic hardship experienced by fishing households. The study recommends interest-free loans to support fisherfolk who are already engaged in small businesses. The provision of supplementary livelihoods could also improve fisherfolk's income and well-being.

## 1. Introduction

The recent increase in global oil prices remain a major challenge for developing countries because of its adverse effects on economic development [1]. The rise in global crude oil prices has led to significant increases in fuel prices worldwide [2]. This has increased living costs and slowed economic growth [2–4]. The Ghanaian economy is highly vulnerable to global oil price changes because oil as a commodity plays a key role in the nation's economic [1,5,6]. Over 90% of the energy utilized in the agricultural sector in Ghana is obtained from oil [6]. Since January 2022, fuel prices have increased by more than 90% [7]; Fig 1.

This study investigates the effects of fuel price hikes on the livelihoods of small-scale coastal fisherfolk (SSF) in Ghana. The study also explores the various coping strategies deployed by fisherfolk in response to fuel price hikes. The SSF is defined as fishing and its related activities, such as processing and trading, operated by individuals and families in coastal communities [8]. Fisherfolk in this study refers to fishers, fish traders and processors, and any other person

**Data Availability Statement:** "All relevant data are within the paper and its Supporting Information files."

**Funding:** The author(s) received no specific funding for this work.

**Competing interests:** The author/s declare that they have no known competing financial interests or personal relationships that could have appeared to influence the work reported in this paper

who depends on fishing and its related activities for their livelihood [9]. Changes in global oil prices could significantly impact agricultural productivity, including fisheries because fuel price is a key driver of inflation in Ghana [1,2];. The rising fuel prices and associated increases in food prices have caused deteriorating effects on household income in developing countries [10].

Across the Global South, the fisheries sub-sector of agriculture remains a major source of livelihood in many coastal communities. An estimated 38.98 million people are into capture fisheries, with another 20.53 million employed in aquaculture [11]. Fish provides more than 3 billion people with a 20% intake of animal proteins, reaching as high as 50% or more in countries such as Bangladesh, Cambodia, the Gambia, Ghana, Indonesia, Sierra Leone, and Sri Lanka [12]. In Ghana, the fisheries sector contributes significantly to employment, income, livelihood, nutritional security, and socioeconomic development [12–14]. The fisheries sector employs about 10% of the active workforce of the population as fishers, processors, boat owners, boat builders, and others in subsidiary jobs. The direct workforce for the industry includes about 140,000 fishermen in the four coastal regions and 4,474 persons engaged and employed in the fishing and aquaculture industry [15].

The increases in fuel prices have the potential to further expose many fishing-based households in developing countries to various forms of vulnerability, including poverty, hunger, and food insecurity. Ghana's oil needs are mainly dependent on imported crude oil, which makes imported crude oil an essential part of the economy, and as such, changing world oil prices is a concern that needs to be examined [6]. The effects of rising global oil prices and its associated

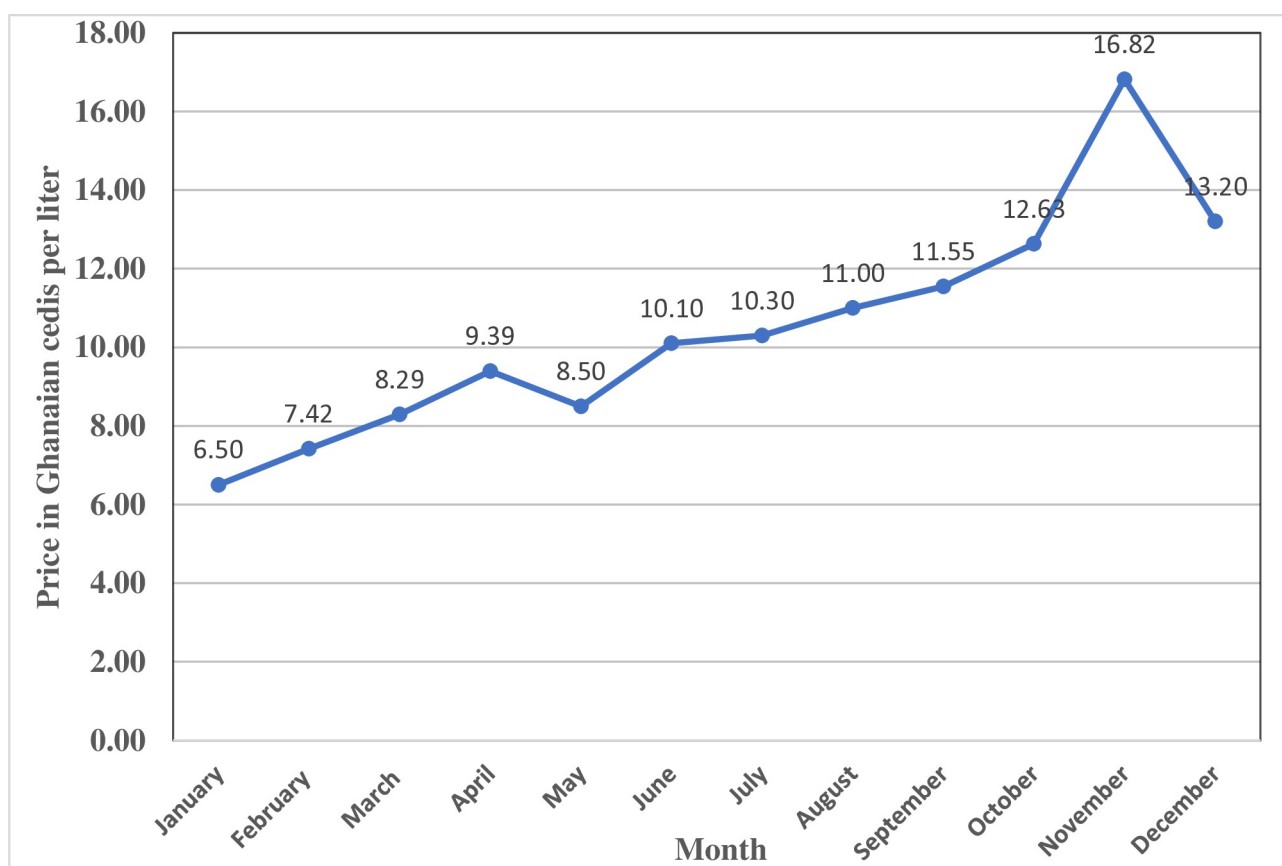

**Fig 1. Changes in fuel price in Ghana (January-December 2022).**

challenges have been analyzed mainly from the macro/national perspectives [3,16]. Studies on the impact of oil prices on various sectors of the economy at the local level in Africa are limited [1,5].

The Sustainable Livelihoods Approach (SLA) is a useful analytical tool for researchers and development practitioners to understand the livelihood trajectory of rural households or individuals at the local level in the midst of global oil price hikes [8,17,18]. The SLA examines the link between individual or household assets and the activities in which households can engage with a given set of assets [19–21]. Access to capital assets (human, social, physical, natural, and financial) and activities are enhanced or hindered by the mediating policies, institutions, and processes (PIPs), as well as social relations, markets, and organizations [22]. The key components of the SLA include the vulnerability context, livelihood capital assets, PIPs, livelihood strategies, and outcomes [23].

The vulnerability context in SLA examines the changes in the external environment and how they affect or influence people's livelihoods. The SLA is applied in this study to investigate artisanal fisherfolk's vulnerabilities and livelihood strategies in the context of global oil price shocks. The global fuel price hikes are considered to have triggered a major shock and stress on the livelihood assets of fisherfolk. Vulnerability comprises seasonality, trends, and shock [22,23]. The trends may include decreasing fish catch, increasing fish prices, and rising foodstuff costs [22]. Shocks may include fuel-price hikes that affect the costs of fishing inputs and market prices for fishery products [1,5,22]. At a household level, the illness or death of a relative and the theft or loss of fishing inputs and other household items are obvious shocks [22].

Other factors that negatively affect fishing households' income and coastal livelihoods include climate change, pandemics, structural change in the economy, and offshore oil production [18,24–26]. Other drivers of change, such as overfishing, which occurs mainly because of illegal fishing activities involving both industrial trawlers and artisanal fisherfolk, has also partly contributed to the dwindling fish catch [9,27]. The industrial trawlers often engage in the use of unauthorized fishing gears, transshipment, and fishing in areas reserved for the SSF [28].

The paper proposes a modified SLA based on insights from studies on rural, coastal development, and energy (Fig 2). The modified SLA pays closer attention to the political economy and show how structural transformations enhance or constrain households' access to capital assets to improve livelihoods and well-being [17,18,21]. Given the macroeconomic context of Ghana, it makes sense to include the stagnated economic transformation as part of a broader process in our understanding of vulnerability [8,18]. Over the past three decades, Ghana has made moderate economic gains and transitioned from low-income to lower middle-income status with a per capita GNI of USD 2,369 [29]. However, the economic growth has not been pro-poor and skewed toward the benefit of only a few people. People working in agricultural-related jobs like fishing and farming remain the poorest [30]. Coastal livelihoods are sustainable when fisherfolk can earn more income, improve food security, and reduce vulnerability, resulting in an overall improvement in the standard of living and well-being.

This paper contributes to the literature on global fuel price hikes and its implications for marine fisheries sustainability by investigating the effects of oil price shocks on fisheries' livelihoods in SSF communities in Ghana. The study considers the impacts and the various coping strategies fisherfolk deploy in response to fuel price shocks. The study contributes to the literature on coastal livelihoods, ocean governance, and natural resources management. Ghana is both an oil importer and exporter; hence, examining the effects of escalating global oil prices on the local fishing industry would interest the government, development practitioners, and policymakers. The results from the study could further help to devise a more specific stimulus package tailored to cater to the unique needs of the sector involved.

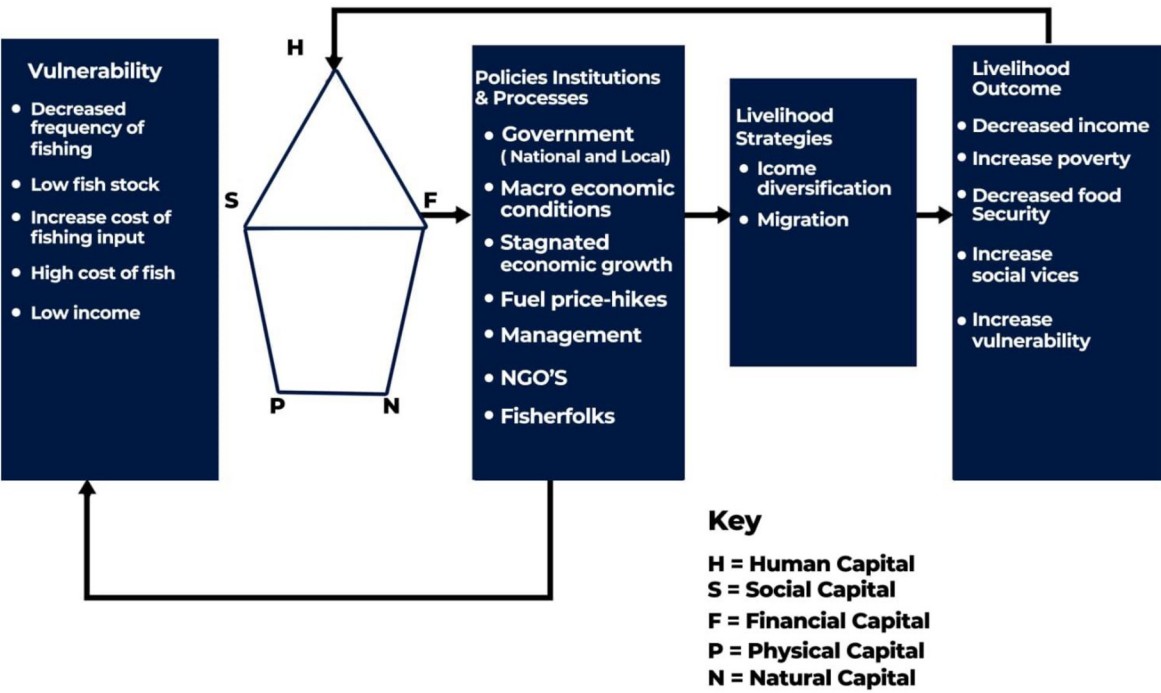

**Fig 2. Analytical framework.**

The study addressed the following two research questions;

1. What have been the effects of fuel price increases on the livelihoods of coastal fisherfolk

2. What are the coping strategies deployed to mitigate these impacts?

The study shows that fisherfolk in the Central Region of Ghana are under high socioeconomic vulnerability because of fuel price hikes. Fisherfolk experienced financial, emotional and psychological shocks due to the high cost of fuel which affected fisheries livelihoods. Dependency on savings, borrowing, petty trading, migration and farming were found to the main coping strategies. However, the various livelihood coping strategies deployed by fisherfolk were not sufficient to ameliorate their economic hardship. The rest of the paper is organized as follows: the next section presents the study areas, and the research methodology description. After that, the results and discussion are presented, highlighting the implications of findings for improving sustainable coastal management, and finally, ends with the conclusion.

## 2. Materials and methods

### 2.1 Study context

In 2007, Ghana discovered oil in commercial quantities, with production beginning in 2010 [9]. The major oil and gas companies operating in Ghana include- Tullow Ghana Ltd, Anadarko Petroleum, KOSMOS Energy, GNPC, Eni, Hess, and Sabre Oil and Gas, collectively referred to as 'Jubilee Partners' [31]. In 2017, the total oil revenue received by the national government was USD362.58 million (GHS1,552.13 m) [32]. This amount increased to USD 723.55 m (GHS3,292.20m) by 2018. By 2019, production had risen to 196,089 barrels per day and is expected to increase to 500,000 by 2025 [33]. The increase in oil production has

contributed to a significant growth rate in Ghana's Gross Domestic Product (GDP). In 2019, Ghana accrued oil revenue totaling USD 938 million, equivalent to nine percent of government revenue [34].

As a net exporter of crude oil, the surge in the global oil price should be positive for Ghana's trade balance. However, the country is highly vulnerable to global oil price shocks [1]. This is because it exports its crude oil and imports a significant share of petroleum products, including petrol and diesel [2,6]. Years of under-investment and neglect have resulted in inadequate local refining, especially at the country's main oil refinery–Tema Oil Refinery. Ghana imports about 95% of oil, with local production accounting for only 5% of about 80% petroleum consumption [1,2].The cost of importing petroleum products increases after global oil prices increase. This is because refined oil serves as inputs and raw materials for domestic [1]. Fuel prices are a vital driver of inflation in Ghana [6]. This means that any hikes in global oil prices feeding through into petroleum product imports will significantly escalate inflation. During periods of increasing oil prices, oil-importing countries reliant on food imports for consumption will experience a surge in their food import bill [6]. The global oil price shock has exposed fundamental structural weaknesses in Ghana's economy [2]. Despite being a net exporter of crude oil, the country has failed to cushion the citizenry against escalating global oil prices. As [1] noted, increasing oil prices generates revenue for oil-exporting countries; however, it also poses a potential threat of causing inflation, recession, low growth, high food prices, and low development for oil-importing countries. This exposes Ghana's economy to shocks from global oil price with adverse repercussions economic development.

The 2015 Sustainable Development Goals (SDGs) (1, 2, & 14) seek to eradicate poverty, hunger, and all forms of food insecurity and malnutrition by 2030, including the sustainable management of marine and coastal ecosystems. However, the challenges to ending poverty, hunger, food insecurity, and malnutrition keep growing [10]. The Russia-Ukraine conflict, the high cost of living, the COVID-19 pandemic, and climate change are affecting the livelihoods of people across the world [24,35]. The COVID-19 pandemic exposed the fragilities in agrifood systems and the inequities driving further increases in world hunger and severe food insecurity [10]. Prior to the recent global oil price surge, the covid 19 pandemic disrupted the global fisheries supply value chains, leading to higher aquatic food prices of which Ghana was no exception [8,24,36]. About 820 million people worldwide suffer from malnutrition, and another 736 million live in extreme poverty, most of which are in developing countries [16]. Ghana is experiencing a significant increasing inequality, uncertainties in fish availability, and unsustainable fisheries management [12]. Demand for fish exceeds the domestic supply, and the shortfall is covered by approximately 200,000 tons of imports [37]. The total export for 2021 was 67,786.03mt (provisional) in 2021 and 37,156.70mt in 2020, a significant 30,629.33mt increase. The country imported 273,382.32mt in 2021 as against 193,226.87mt in 2020, which saw an increase of 80,155.45mt [15].

Ghana has a coastline of 550km along the Gulf of Guinea. The fisheries sector consists of marine capture fisheries, inland fisheries, and aquaculture. The marine fisheries sector comprises four subsectors: artisanal or small-scale, semi-industrial or inshore, industrial, and tuna fisheries [15]. Marine fishing is an important traditional economic activity of the coastal communities, contributing over 80 percent of the total fish catch [38–40]. Marine fisheries resources include small pelagic, large pelagic (Tunas), and demersal species. The most important small pelagic include round sardinellas, flat sardinellas, anchovy, chub, and horse mackerels, contributing 60–70% of total marine fish production [15]. The small pelagic contributes immensely to food and nutrition security. The most important demersal species include sea breams, red snappers, groupers, grunts, croakers, cephalopods, and shrimps and, together with the large pelagic (Bigeye, Yellowfin, and Skipjack), contribute significantly to foreign

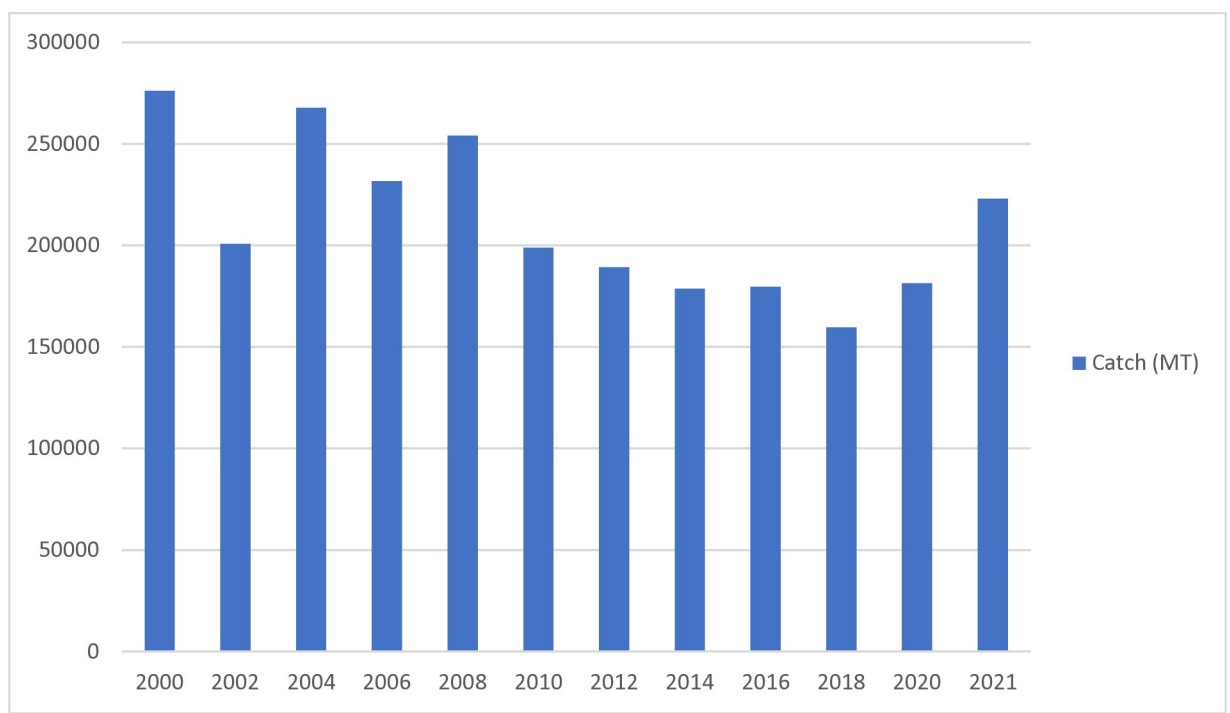

**Fig 3. Total marine fish production by artisanal canoe 2000–2021 (metric tons).**

exchange earnings [15]. The total domestic production by artisanal canoe in 2021 amounted to 223,088.61mt compared to 181498.81mt in 2020 (Fig 3). Overall, the marine sector recorded 393,970.01mt in 2021 against 297,976mt in 2020, representing a 32% increase [15].

Two coastal communities located in the Central Region were selected for this study. Winneba is a coastal town in southern Ghana. It is located in the Central Region and serves as the capital of the Effutu Municipal. Winneba lies on latitude 5˚22'01.5"N and longitude 0˚ 37'52.2"W along the Gulf of Guinea. Winneba is situated 56 km west of Accra, Ghana's capital city, and 140 km from Cape Coast [41]. Winneba is a vibrant fishing area with five landing sites, 4270 fishermen, and 436 canoes [37]. The main fishing grounds include Eyipey, Winneba, Esuakyir, Warabeba, and Akosua Village. Mackerel, kingfish, tuna, sea bream, and herring are some fish species caught here [37]. Besides marine fishing, the Muni Lagoon is an important fishing ground for most fisherfolk. Three finfish and eight shellfish species are found in the lagoon. The dominant species is black chin tilapia, *Sarotherodon Melanotheron* (Cichlidae) [42]. The men are responsible for fishing, while the women are involved in fish processing, trading, and marketing.

Apam is a coastal town located along the Atlantic Ocean in Ghana. It is the capital of the Gomoa West District in the Central Region of Ghana. It is approximately 68 km east of the Central Regional capital, Cape Coast, and about 69 km southwest of Accra [43]. It lies on the coordinates 5˚18'17.91"N, 0˚42'34.89"W, and 5˚14'47.33"N, 0˚46'35.27"W. The main occupations of the people in the district are subsistence agriculture and fishing [37]. According to [37], ten landing sites, 4,062 fishermen, and 298 canoes are registered in the community to support the fishing industry. The men engage in fishing, whilst the women undertake processing, selling of processed and fresh fish. In addition, the sector also indirectly supports the livelihoods of boat builders, food vendors, and transport operators within the community. The main fishing season in Apam is between August and December. Some of the commonly caught fish include threadfin (*Galeoides decadactylus*) and burrito (*Brachydeuterus auritus)* [37].

## 2.2 Research methodology

**2.2.1 Data collection.** The study used a mixed-method approach to investigate the effects of fuel price hikes on the livelihoods of coastal fisherfolk. The empirical data consists of a questionnaire survey of 320 SSF-dependent households and 20 interviews with stakeholders in the fisheries sector including fisherfolk, government officials and a representative from NGO. The use of both quantitative and qualitative approaches provided a better comprehension of the research problem and facilitated cross-validation and triangulation. The surveys and interviews were conducted between July and August 2022 in the communities of Winneba and Apam in the Central Region of Ghana. The data was gathered in the language the respondent best understood after obtaining informed consent. The languages used included English, Fante, and Twi. The questionnaire and interview guide were designed in English; however, when a respondent did not understand the questions, the local languages mentioned above were used to explain the questions. To ensure interviewer reliability for consistent data collection, a common standard for selecting and training the research assistants was set. All the research assistants had fundamentals in conducting research and fluency in English, Fanti, and Twi languages. The research assistants were trained in the data collection processes and had sufficient knowledge of the issues under investigation.

The household questionnaire covered three broad thematic areas: socioeconomic and demographic information, effects of fuel price hikes, and coping mechanisms. The questionnaire included closed-ended, open-ended, and multiple-choice questions. The open-ended questions allowed the respondents to freely talk about their experiences and knowledge relevant to the local fishing industry and other relevant issues concerning the topic under investigation. The survey lasted between 50 minutes to an hour on average per person. The major themes in the interview guide was the socio-demographic information on respondents, fuel prices and fishing operations, and the coping strategies deployed to mitigate the effects.

**2.2.2 Sampling method and sample size.** Quantitative and qualitative sample sizes were determined using probability and non-probability sampling procedures. Simple random and purposive sampling techniques were employed in selecting respondents for the household survey and interviews. According to the 2016 national canoe frame survey, there were 4270 fishermen in Winneba and 4062 in Apam. The number of fishermen in the study communities was similar during the survey. Information on the population of fishermen was sourced from the Chief fisherman to constitute the sample frame. A total of 320 fisherfolk were engaged in the household survey, including 170 randomly selected fishers engaged in small-scale commercial fishing from Winneba and 150 from Apam. Random sampling enables each individual in the population to have an equal chance of being selected [44]. Based on information provided by the chief fishermen that showed the presence 4270 fishermen in Winneba and 4062 in Apam. The sample size was therefore determined using Yamane formula:

$S_n = \frac{N}{1+N(e)^2}$

Where:

Sn = sample size

N = population size (8,332)

e = margin of error

$S_n = \frac{8332}{1+8332(0.05)^2}$

$= \frac{8332}{21.83}$

= 381.67

= 382 approximately

The sample sizes were proportional to the population of each area, with Winneba being slightly larger (Table 1). However, 320 (84 percent) of the 382 respondents provided usable

Table 1. Sample structure for selecting fisherfolk.

| Study areas | Population | Proportion | Sample size |
|---|---|---|---|
| Winneba | 4,270 | 51% | 195 |
| Apam | 4,062 | 49% | 187 |
| **Total** | **8332** | **100** | **382** |

information for the study. The remaining 14% either did not complete the survey or declined to do so. Even though not up to the required sample size, the data is still large enough to ensure statistical validity for comparison between the two areas. The sizes are within a reasonable range to detect significant differences or trends in the population without introducing major biases.

The purposive sampling technique was used to select community leaders, government officials, and NGO representatives. The in-depth semi-structured interviews (N = 20) were conducted with community leaders comprising chief fishermen (n = 2), chief fish traders (n = 3), chief fish processors (n = 3), canoe owners (n = 7), government officials (n = 4), and NGO official (n = 1) to gain insights into the experiences of fisherfolk concerning fuel price hikes. These stakeholders were considered to have the requisite knowledge and experience with fishing and other related activities in the studied communities. The goal here is to acquire an in-depth understanding of the subject of the study from participants viewpoint. The sample size was determined upon saturation point (see Hay 2010). These sampling techniques are efficient data-gathering methods and have been used by several authors to investigate fisheries livelihoods and coastal development in West Africa [26,27,45].

**2.2.3 Data analysis.** SPSS Version 23.0 and Excel were used to analyze the results from the household surveys.

The socio-demographic characteristics of the respondents were presented in frequency tables. The effects of the fuel price hike, and coping strategies were presented in tables and charts. The interviews were audio-recorded with the respondents' permission and later translated verbatim. The transcribed data was manually coded and organized into relevant themes based on the study's objectives and components of SLA used in the study. Qualitative content analysis technique was used to interpret meaning from the content of text data [46]. The author read through the transcripts severally to fully comprehend responses given by research participants and highlighted particular words from the text that appear to capture critical thoughts and answer the research questions. Selected narratives from the in-depth interviews were then presented as direct quotations to illustrate key findings. Direct field observation of the coastal environments and fishing activities further provided a depth of information. Participant observation was also conducted at the fish markets and landing beaches. Information related to the organization of fisheries and other related activities, such as trading and processing, were recorded. These observations provided insights into the social and economic environments where fishers perform daily economic activities and related social functions. Secondary data was collected from published and non-published reports, online newspaper articles, and internet sources to support the analysis of the study.

## 3.Results

Table 2 presents an overview of selected socio-demographic features of the two study communities. The study communities in Apam recorded the highest household number, with 33.3% of respondents indicating they have more than 10 members in their household. Apam has the highest number of fishers with no formal education (43.2%). Across the study areas, the high

**Table 2. Characteristics of the study sites and persons interviewed (n) = 320.**

| Variable Value | | APAM | WINNEBA |
|---|---|---|---|
| AGE | BELOW 20 | 3.8 | 5.9 |
| | 20–39 YEARS | 44.2 | 43.1 |
| | 40–59 YEARS | 38.5 | 37.3 |
| | ABOVE 60 | 13.5 | 13.7 |
| MARITAL STATUS | | | |
| | SINGLE | 17.3 | 29.4 |
| | MARRIED | 59.6 | 45.1 |
| | DIVORCED | 13.5 | 13.7 |
| | SEPARATED | 9.6 | 11.8 |
| EDUCATIONAL LEVEL | | | |
| | NO FORMAL EDUCATION | 43.2 | 28.8 |
| | ELEMENTARY | 37.2 | 50.0 |
| | SECONDARY | 17.6 | 21.2 |
| | TERTIARY | 0.0 | 0.0 |
| | TECHNICAL TRAINING | 2.0 | 0.0 |
| TYPE OF FISHER | | | |
| | CAPTAIN | 25.0 | 21.6 |
| | CREW MEMBER | 40.4 | 41.1 |
| | BOAT/CANOE OWNER | 34.6 | 37.3 |
| NUMBER OF PEOPLE IN YOUR HOUSEHOLD | LESS THAN 5 | 13.4 | 15.7 |
| | 6–10 | 61.5 | 51.0 |
| | MORE THAN 10 | 25.1 | 25.1 |
| WHAT IS YOUR AVERAGE MONTHLY INCOME FROM FISHING? | 0–199 | 0.0 | 0.0 |
| | 200–399 | 5.7 | 3.9 |
| | 400–599 | 26.9 | 27.5 |
| | 600–799 | 23.2 | 23.5 |
| | MORE THAN 800 | 44.2 | 45.1 |
| WHAT IS YOUR AVERAGE MONTHLY INCOME FROM OTHER INCOME-GENERATING ACTIVITIES? | 0–199 | 76.9 | 78.4 |
| | 200–399 | 17.4 | 15.7 |
| | 400–599 | 5.7 | 2.0 |
| | 600–799 | 0.0 | 0.0 |
| | MORE THAN 800 | 0.0 | 2.0 |
| AVERAGE FUEL COST /PER FISHING TRIP | Before 2022 | GHS350 | GHS350 |
| | After 2022 | GHS550 | GHS550 |
| TOP TWO PROBLEMS IN THE COMMUNITY AT THE MOMENT | High cost of living<br>Unemployment | 100.0<br>50.0 | 100.0<br>70.0 |

Exchange rate: USD 1 = GHS 9 as of August 2022.

**Table 3. Key survey results.**

| Variable Value | | APAM | WINNEBA |
|---|---|---|---|
| Purpose of catch | Commercial | 100 | 100 |
| | Domestic | - | - |
| | | | |
| Catch quantity changes (over the past 5 years) | Increase | 3.0 | 2.0 |
| | No change | 5.0 | 4.0 |
| | Decrease | 92.0 | 94.0 |
| Income changes (over the past 5 years) | Higher | 2.0 | 3.0 |
| | No change | 6.0 | 4.0 |
| | Lower | 92.0 | 93.0 |
| | | | |
| How long have you been fishing? | Less than 5 years | 3.8 | 5.9 |
| | 5 to 10 years | 13.5 | 15.7 |
| | 11 to 15 years | 36.5 | 31.4 |
| | Over 15 years | 46.2 | 47.0 |
| Do you have other jobs besides fishing? | Yes | 5.7 | 21.6 |
| | No | 94.2 | 78.5 |
| | | | |
| Has the increase in fuel prices in any way affected your business operation? | | | |
| | Yes | 100 | 98.1 |
| | No | 0.0 | 1.9 |

cost of living was the topmost perceived problem (Table 2). Research participants indicated that the increase in fuel prices was the major cause of the high cost of living. The majority of the fisherfolk in the study areas have no alternative work apart from fishing (Table 3).

## 3.1 Perceived effects of high fuel price on fisheries livelihoods

Four main issues were identified concerning how increased fuel prices affected the livelihoods and well-being of coastal fisherfolk. These include an increase in expenses, reduced frequency of fishing, an upsurge in social vices, and declining small-scale fisheries opportunities (Fig 4).

The results from the household survey, coupled with the interviews, revealed that there has been a substantial increase in expenses concerning fishing operations. The Government of Ghana distributes premix fuel to coastal fishing communities at a much-subsidized price. Premix fuel subsidy was introduced in Ghana's small-scale fisheries in the 1990s to deal with the high cost of fuel and reduce the socioeconomic burden on the poor and vulnerable fisherfolk in coastal communities [47,48]. The subsidized premix fuel, which was sold at GHS10 per gallon, has been increased to GHS25 per gallon because of the rising oil prices [49]. The majority of fisherfolk reported that the increase in fuel prices has negatively affected the supply of premix fuel. Fishers have to wait for several weeks and months without premix fuel supply, and the only option is to buy expensive petrol for fishing activities:

> *Since the price of fuel started increasing, the supply of premix fuel has not been frequent. The price of the premix fuel has also increased, which has rendered most of us unable to undertake our fishing activities, especially those who embark on fishing activities at night.*

(Canoe owner, Apam)

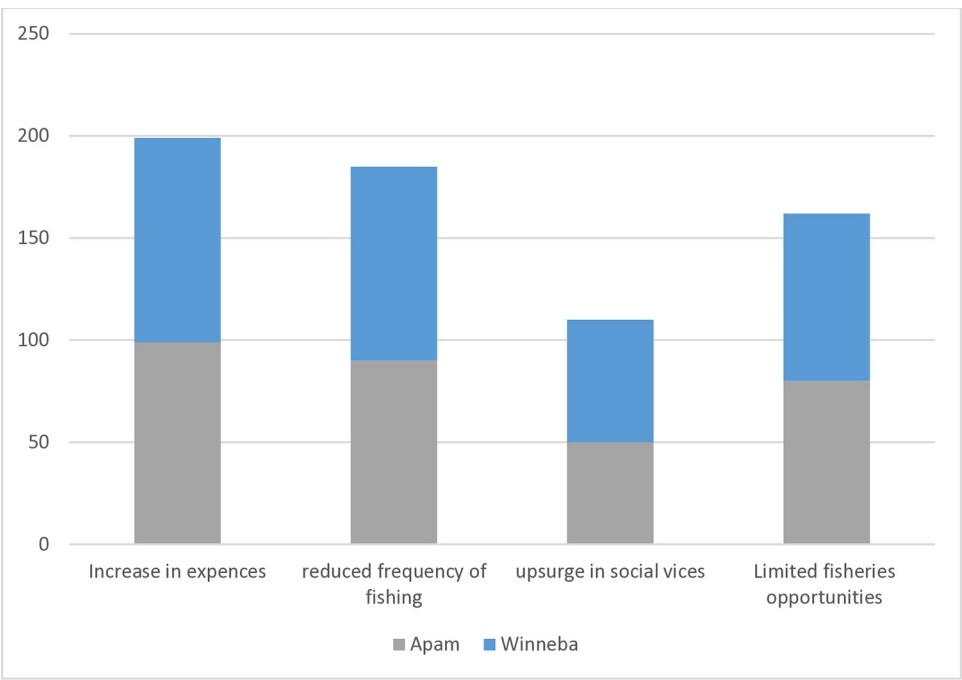

**Fig 4. Perceived effects of high fuel price on fisherfolk livelihoods.**

*The rise in the price of petrol has brought a lot of negative effects on the fishermen. Those who sell the petrol to us don't even get some to buy for us to purchase from them. This has caused an increase in the price of the premix fuel, because if you want to buy the fuel unless you resort to third parties rather than the normal channel of buying it.*

(Chief Fisher, Apam)

Fisherfolk emphasized that even though the supply of the premix fuel was irregular, the recent fuel price hikes have worsened the situation. The majority of the fisherfolk reported that the increase in fuel prices has reduced the frequency of fishing operations, resulting in low catch and income. Interviewed canoe owners reported spending more on fuel and food for their crew members. According to one of the interviewed canoe owners before 2022 he spends between GHS350 to GHS400 on fuel per fishing trip with boat engine capacity of 40hp. They were able to spend several hours at sea with decent catch between 200-350kg of fish. However, when the fuel prices started increasing in 2022, he is unable to buy enough fuel for fishing which has affected the quantity of fish catch. He currently spends more than GHS 500 on fuel per fishing trip and catches between 100–150 kg. This has further reduced the number of fishing trips as most canoe owners run at a loss because of the persistent decrease in the fish catch. According to him, the fuel price hikes has reduced the profit margin of canoe owners with most fishing businesses on the verge of collapsing. Before 2022, when the cost of fuel was low, fishers often embark on fishing almost every day. During this period fishers reported significant fish catch with good profit margin.

The following extracts from the interviews with fishers aptly speak to the findings presented above:

*I have lost about GHS2000 this month. Anytime there is an increase in fuel prices, fishermen suffer a lot. The high fuel prices have affected everything. A small gallon of petrol is sold at*

*GHS40 while the bigger gallon is over GHS250. Two of my canoes have been packed because of the high fuel price. I do not have any additional work besides fishing.*

(Canoe owner and Fisher, Winneba)

*The increases in fuel price have impacted us negatively in many ways. Most of us now incur losses as a result of the increment. We used to fish five times during the week; however, because of the high fuel cost, it has been reduced to only two fishing trips. The quantity of fuel determines the duration of fishing. Our canoe owner says he cannot afford more fuel because of the higher prices.*

(Fisher, Apam)

Responses generated from the interviews with local fisherfolk—fishers, canoe owners, and local leaders revealed that increases in fuel prices have a negative implication on fishing households' financial capital development. Fuel constitutes a substantial component of the cost of fishing; globally, about 30–50% of fishing expenditure is associated with fuel [50]. In developing countries, fuel constitutes about 50% of fishing expenditure within small-scale fisheries [35,50].

The increases in fuel prices affect other fishery-related businesses, such as processing and trading. Interview with some of the fishmongers and traders indicates that the increase in fuel prices has affected the quantity of fish supply and the price of fish. This point was echoed by the fish processors:

*The increase in fuel prices has led to the high cost of fish, which has brought us economic hardship. We are unable to buy large quantities of fish as we did previously. This has affected our income, and we struggle to care for our families.*

(Fish processor and trader, Winneba)

*The inflow of petrol to the coast is not frequent and this has compelled many fishermen to buy fuel from 'middlemen' at expensive cost. The prices of fuel have caused an increase in the prices of fish. About 20 years ago, when the fisheries business was good, l was able to establish a small trading shop. However, in recent times, we barely make any profit from the fishery business to support our families and invest in other businesses. I'm appealing to the government to reduce the price of fuel and increase the supply for the fishermen to facilitate their work.*

(Fish processor and trader, Apam)

The fishers reported that they sell fish at a much higher price because of increases in the prices of fuel:

*We sell fish to the women, and they complain it is expensive, but we have to pay for the high fuel cost. The rise in the price of petrol has affected us in so many ways, and those who even sell petrol to us do not even get some to buy for us to purchase, which has worsened the price of the premix fuel. We used to buy a drum/gallon of premix fuel at GHS700, but now it is sold at GHS1200-1300.*

(Fisher, Apam)

Results from the household surveys and interviews show that fisherfolk are currently undergoing severe economic hardship. The increased expenditure on fishing inputs such as fuel

remains a major financial burden on fishing households. A recent study by [51] found that a box of fish sold at GHS 250 is now sold between GHS450 and GHS500. The increase in fuel prices has significantly led to increases in inflation, with food prices being the most affected [52]. According to the [52], prices of food items such as cereal (rice, corn, millet, grains, wheat, sorghum), meat, fish and other seafood, milk, dairy products, eggs, and cooking oil increased significantly.

Many fishing households reported that their children are not able to go to school because of a lack of money to cater for their needs:

> *I find it difficult to cater to my children's education. Though my husband is working, he does not earn much to take care of our children, not to mention housekeeping money from him for our feeding. I mostly pay for my children's school fees. However, because of financial problems, they must stay home for some time before returning to school. Some of us barely eat once in a day.*

(Fish processor and trader, Apam).

Another participant remarked:

> *Our children are suffering because there has been a frequent rise in the price of fuel. We are suffering; the hardship is too much for us. They told us we would not pay school fees, but we were paying more than the school fees. Some of my children have dropped out of school because of my inability to pay their school fees. Many of us now rely on our mothers and buyers for survival. I have lost much money as a result of the high cost of fuel, and this has compelled most of us to resort to borrowing.*

Another social problem emerging from the fuel price hikes is the upsurge of social vices and increased community tension. Many fishing households reported that their money, foodstuffs, and other personal properties were stolen. Many respondents revealed that the upsurge in social vices has led to considerable frustration, stress, and anxiety in the community:

> *Apart from financial problems, I go through psychological stress because I get traumatized and think a lot about how I will survive since I do not have any other work besides fish processing. Many have lost their jobs and livelihoods, and there has been an increase in social vices in the community.*

(Fisher processor and trader, Apam)
Another fisher opined:

> *The economic hardship has made some youth in the community resort to illegal ways of getting money to cater for themselves and their family. This has led to robbery in our neighborhoods. Since this town has no jobs, we are appealing to the government to create jobs for us.*

The empirical findings from this study suggest that the fuel price hikes resulted in the loss of income and heightened social tension due to increased social vices. This negatively impacts the human and physical capital development in coastal communities. Increased anxiety and financial stress affect the well-being and health of fisherfolk in coastal communities.

## Livelihood coping strategies

The livelihood strategies reported by fisherfolk include dependency on savings, borrowing, petty trading, and farming. 60% of surveyed fisherfolk in Apam depended on their savings and

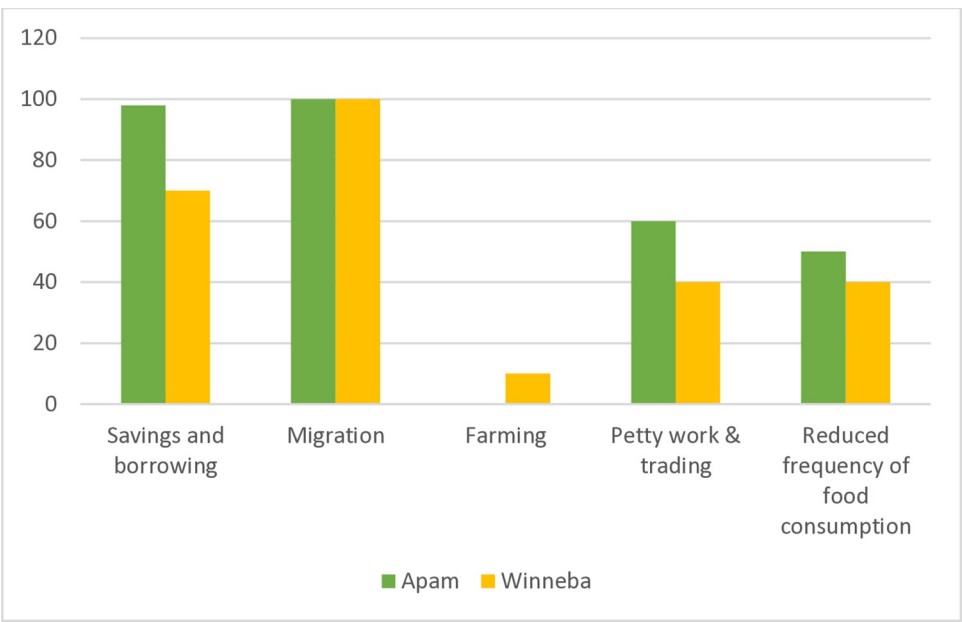

**Fig 5. Coping strategies (based on fishers' multiple-choice responses, % of fishers;' n = 320).**

borrowing to care for their families, while 50% of fisherfolk in Winneba depended on their savings and borrowing (Fig 5).

The fishers borrow money from canoe owners, friends and families, and fish traders. They also sometimes buy foodstuffs on credit from fish traders, processors, and other market women. Most fishers reported that they defray their debt with fish during the major fishing season:

> I borrow money from the fishmongers, and l always pay back with fish. Sometimes, we buy food and other commodities on credit, hoping to repay them during the bumper harvest season. However, a major challenge is that because we are not getting a lot of fish during major fishing seasons, it becomes difficult to pay back our debt. This has resulted in many people refusing to lend us money and food items.

(Fisher, Winneba)

Some fisherfolk also engaged in petty trading, such as selling coconut and used clothing. Others engaged in handicrafts such as mat weaving;

> I save from the little I get when I sell the fish and engage in other trading activities that involve selling used clothing to help me care for my children because I am a single parent.

(Fish processor and trader, Apam)

In interviews, some fish traders mentioned that the lack of capital to expand the business remains a major challenge to sustain their operation. Moreover, the demand for items is not encouraging, making it difficult to make profit.

Farming is practiced by just 10% of fisherfolk in Winneba. Some of the common crops cultivated are maize and cassava. In Apam, no fisherfolk reported any farming activity. Another major important coping strategy reported by fisherfolk is migration. Fishers in both study communities reported that they regularly travel to fish in other coastal communities:

*I travelled to Weija to fish, we sometimes do not get a good catch, and the fuel cost affects us, but we do not have any job apart from fishing. We also sometimes travel to fish at Elmina and Cape Coast to increase our catch.*

(Fisher, Winneba)

There are no jobs in this area, the only job we have is fishing. We regularly made enquiries from other coastal areas to ascertain where there is abundant catch. When we hear of abundant catch in other areas, I mobilize my crew and travel to those areas to catch fish.

(Canoe owner, Apam)

Migrating to fish is an important coping/adaptation strategy among fisherfolk in Ghana [9,45].

Other coping strategies reported include reducing the frequency and quantity of food consumption. Fisherfolk reported that the reduction in the frequency and quantity of food consumed preceded the fuel price hikes. However, it became more pronounced since fuel prices started increasing. 50% of respondents in Apam indicated that they have reduced the frequency of food consumption whilst 40% of respondents in Winneba indicated that they have also reduced the frequency of food consumption.

## 4. Discussion

This paper sought to show how recent fuel price increases affect small-scale fisheries' livelihoods and marine sustainability. The study also explores the various coping strategies deployed by fisherfolk in response to rising fuel prices. Increases in expenses, decreased fishing frequency, upsurges in social vices, and declining small-scale fisheries opportunities were found to be the main effects of high fuel prices on fisheries livelihoods. Fuel price hikes have negatively impacted fishing households' financial, human, and social capital assets.

The financial capital assets of coastal fishing communities have deteriorated because of the fuel price increases. The high fuel cost has led to decreased fishing frequency resulting in low fish stock, and income (Table 4). Households that entirely depend on fishing and its related activities are most affected as they do not have any viable alternative sources of income (Table 3). Households with fewer livelihood options are most vulnerable to external shocks and are susceptible to falling back into poverty [8,53]. The increases in fuel prices have also resulted in high cost of fish at the various landing sites. This has brought financial burden on women, as they are unable to buy large quantities of fish for processing and trading. Previous studies show that women are among the most vulnerable in coastal fishing communities due to their limited ownership of fishing assets and participation in fishing work [54,55]. Fishers

**Table 4. Impacts of the high fuel cost on livelihood assets of fisherfolk.**

| Capital asset | Perceived influence of high cost of fuel on livelihood assets |
|---|---|
| Natural capital | Limited fishing operation and low fish stock undermines traditional livelihoods. |
| Human capital | Reduced fish consumption undermines nutritional and food security; frustration, anger, and anxiety affect the health and well-being of fishing households. No provision for training or capacity building for fisherfolk to undertake alternate livelihoods |
| Financial capital | High cost of fuel, increased cost of living, no additional sources of finances to support daily subsistence, and low savings affect investment into other economic sectors. |
| Physical capital | Loss of material properties due to theft, limited ownership of productive assets such as canoes and nets. |
| Social capital | Weakening social capital, increases social vices erodes communities' collective action, and increases social tension. |

consider fishing to be an important occupation that provides their primary source of income and considering that there are no viable alternative income-generating activities, the high fuel prices impose harsh economic conditions and undermine traditional livelihoods. The small-scale fisheries in West Africa have suffered from rapidly declining marine fish landings, resulting in decreased income and severe economic hardships [55–57]. The decrease in income and declining small-scale fisheries opportunities has led to disruption in the education of fisherfolk children. Fisherfolk are unable to pay their children school fees leading to school dropouts or temporarily stopping school.

The findings from this study also show that there has been an upsurge in social vices. The upsurge in social vices has led to high levels of frustration, anxiety, and stress among the fisherfolk. Fisherfolk are in fear and panic because of the persistent loss of personal properties. As fishing communities are recovering from the fear, panic, and anxiety induced by the COVID-19 pandemic on their livelihoods [24,36], the recent fuel price hikes has worsened living conditions of fisherfolk. The findings from this study reinforces the need to pay more attention to the psychological and mental health needs of coastal fisherfolk. As noted by [8] village-based livelihood analysis has overly focused on developing financial and social capital with little emphasis on the well-being and health component of rural folks. Without good health, fisherfolk cannot perform their daily economic activities. Increased financial stress and anxiety emanating from fuel price shocks could trigger other diseases that further impoverish the health status of fisherfolk.

Despite the uncertainties that fisherfolk have to face, the increases in fuel prices can indirectly contribute towards the restoration of depleted marine fish stock through a reduction in fishing pressure and overcapacity. Several scholars have concluded that Ghana's fisheries resources are overfished and overexploited [12,58,59]. The rise in fuel prices, coupled with the increases in the prices of subsidized premix fuel, can translate into positive marine ecological gains in the long term despite the current socioeconomic hardship experienced by coastal fishing communities [50].

The empirical findings from the study revealed that fisherfolk engage in both marine and non -marine coping strategies to mitigate against the socioeconomic hardship induced by fuel price hikes. The marine -based coping strategies include traveling to fish in other coastal communities. The non- marine based coping strategies included farming and petty trading. However, the various livelihood strategies deployed by fisherfolk were not enough to ameliorate their economic hardship. Livelihood strategies are constantly being revised to cope with contemporary trends of vulnerabilities [18]. Individuals and households explore new forms of livelihoods by expanding their non-fisheries-related income sources while maintaining their base in fisheries [60]. The findings show that fisherfolk's alternative sources of income could hardly sustain them as majority depends solely on the fisheries (Table 3).

Understanding how people succeed or fail in sustaining their livelihoods in the face of global oil price shocks can help design policies and interventions to assist peoples' coping strategies [1,22]. This study provides two implications for SSF-related policies in coastal Ghana: The introduction of social intervention programs and supplementary livelihoods programs. There should be better-coordinated management strategies by the national government and concerned stakeholders to provide incentives that contribute directly to improving coastal fisherfolk livelihood conditions. In this regard, some fund can be set aside and deposited into local welfare funds to compensate vulnerable fishing households during global oil price shocks. It is anticipated that this type of compensation package and incentives can allay coastal communities' fears and worries about persistent fuel price increases. Cash and non-cash payments are increasingly used in marine resource management to protect fishery stocks and compensate for income loss [61,62]. For instance, in Bangladesh and the Brazilian Amazon, fishers

receive food relief items, direct cash transfers, and support for alternative income-generating activities [62,63]. Involved government agencies should demonstrate more political will to improve human and financial capital development in coastal fishing communities by -improving access to education, credit, and healthcare facilities. Specific programs that target training, improved access to credit, and education on savings mechanisms would improve coastal communities' human and financial status. The findings show some moderate number of fisherfolk are already engaged in various petty trading activities The government can therefore introduce interest-free loans to support fisherfolk who are already engaged in small businesses. Providing business training opportunities could also improve the management of finances. The provision of supplementary livelihoods could improve fisherfolk's income and well-being. The government must create the enabling conditions -by providing the necessary facilities and logistics to train and enhance the capacity of fisherfolk. The study recommends aquaculture training for coastal fishing communities. The training on best practices in small-scale aquaculture will be relevant to women in the coastal communities, especially those directly connected to the fishing industry, as women are not allowed to go to sea in traditional Ghanaian fisheries [54].

## 5. Conclusion

This paper investigated the effects of fuel price increases on the small-scale fishery's livelihoods and marine sustainability in coastal communities in Ghana. The paper applied the SLA as a theoretical framework to analyze artisanal fisherfolk's vulnerability, capital assets, livelihood strategies, and outcomes. Increase in expenses, reduced frequency of fishing, an upsurge in social vices, and declining small-scale fisheries opportunities were found to be the main effects of fuel price hikes on fisherfolk livelihoods. The results reveal that fisherfolk experienced financial, economic, emotional and psychological shocks due to the high cost of fuel. Dependency on savings, borrowing, petty trading, migration and farming were found to the main coping strategies. However, the various livelihood coping strategies deployed by fisherfolk were not sufficient to ameliorate their economic hardship. The results show that changes in fuel prices have implications on marine sustainability. Prior to the fuel price changes in 2022, fisherfolk were able to buy more fuel and spent more time fishing contributing to overfishing and depletion of fish stock. However, the recent increases in the cost of fish production due to fuel price hikes coupled with the reduction in frequency of fishing trips can contribute to marine sustainability in the long -term. The reduction in fishing pressure together with stringent enforcement of laws on illegal fishing activities could yield positive ecological outcomes.

The government and other concerned stakeholders must expedite action on renewable energy programs to diversify the country's energy mix and improve energy security. There is a need for effective policies that can mitigate global oil price shocks. The Tema oil refinery could be revamped and equipped to process local crude oil to meet the high demand for petroleum products. This will help reduce economic pressures from global oil price shocks in the national and local communities.

The present study focused mainly on two fishing communities in the Central Region of Ghana. Thus, future studies should consider multiple case study sites of coastal areas using qualitative and quantitative approaches.

## Supporting information

**S1 Table. Sociodemographic information of respondents.**
(DOCX)

**S2 Table. Changes in fisheries livelihoods.**
(DOCX)

**S1 Fig. Relative frequency of impacts of high cost of fuel on fisherfolk livelihoods (n = 320).**
(TIF)

**S2 Fig. Relative frequency of the distribution of fisherfolk coping mechanism (n = 320).**
(TIF)

**S1 File. Fisher comment on high cost of fuel.**
(DOCX)

**S2 File. Fisher complained about the lack of transparency in the sharing of fuel.**
(DOCX)

**S3 File. Mismanagement and elite capture of fuel distribution.**
(DOCX)

**S4 File. Shortage of fuel affecting fishing activities.**
(DOCX)

**S5 File. Borrowing as coping mechanism.**
(DOCX)

**S6 File. Working as labourer to support livelihood.**
(DOCX)

## Acknowledgments

The author wishes to thank all the research assistants and research participants for making this work possible.

## Author Contributions

**Conceptualization:** Victor Owusu.

**Data curation:** Victor Owusu.

**Formal analysis:** Victor Owusu.

**Investigation:** Victor Owusu.

**Methodology:** Victor Owusu.

**Writing – original draft:** Victor Owusu.

**Writing – review & editing:** Victor Owusu.

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
