## [Decision Letter · Decision Letter 0]

13 Aug 2024

PONE-D-23-38320Effect of rising fuel prices on small-scale fisheries livelihoods and marine sustainability in Ghana.PLOS ONE

Dear Dr. Owusu,

Thank you for submitting your manuscript to PLOS ONE. After careful consideration, we feel that it has merit but does not fully meet PLOS ONE’s publication criteria as it currently stands. Therefore, we invite you to submit a revised version of the manuscript that addresses the points raised during the review process.

**ACADEMIC EDITOR: **

Please see comments below

We look forward to receiving your revised manuscript.

Kind regards,

Charles Odilichukwu R. Okpala, PhD

Academic Editor

PLOS ONE

3. You indicated that ethical approval was not necessary for your study. We understand that the framework for ethical oversight requirements for studies of this type may differ depending on the setting and we would appreciate some further clarification regarding your research. Could you please provide further details on why your study is exempt from the need for approval and confirmation from your institutional review board or research ethics committee (e.g., in the form of a letter or email correspondence) that ethics review was not necessary for this study? Please include a copy of the correspondence as an ""Other"" file.

5. PLOS requires an ORCID iD for the corresponding author in Editorial Manager on papers submitted after December 6th, 2016. Please ensure that you have an ORCID iD and that it is validated in Editorial Manager. To do this, go to ‘Update my Information’ (in the upper left-hand corner of the main menu), and click on the Fetch/Validate link next to the ORCID field. This will take you to the ORCID site and allow you to create a new iD or authenticate a pre-existing iD in Editorial Manager. Please see the following video for instructions on linking an ORCID iD to your Editorial Manager account: https://www.youtube.com/watch?v=_xcclfuvtxQ.

6. Please amend your manuscript to include your abstract after the title page.

7. We note that Figure 4 in your submission contain [map/satellite] images which may be copyrighted. All PLOS content is published under the Creative Commons Attribution License (CC BY 4.0), which means that the manuscript, images, and Supporting Information files will be freely available online, and any third party is permitted to access, download, copy, distribute, and use these materials in any way, even commercially, with proper attribution. For these reasons, we cannot publish previously copyrighted maps or satellite images created using proprietary data, such as Google software (Google Maps, Street View, and Earth). For more information, see our copyright guidelines: http://journals.plos.org/plosone/s/licenses-and-copyright.

1. You may seek permission from the original copyright holder of Figure 4 to publish the content specifically under the CC BY 4.0 license. 

Additional Editor Comments:

It is a well written piece of scholarly work. I encourage authors to attend to concerns raised by one of the reviewers.

Look forward to your revised manuscript.

Reviewers' comments:

Reviewer's Responses to Questions

**Comments to the Author**

1. Is the manuscript technically sound, and do the data support the conclusions?

Reviewer #1: Yes

Reviewer #2: Partly

2. Has the statistical analysis been performed appropriately and rigorously? 

Reviewer #1: Yes

Reviewer #2: No

3. Have the authors made all data underlying the findings in their manuscript fully available?

Reviewer #1: Yes

Reviewer #2: Yes

4. Is the manuscript presented in an intelligible fashion and written in standard English?

Reviewer #1: Yes

Reviewer #2: Yes

5. Review Comments to the Author

Reviewer #1: The paper is well-written and adds to the existing literature on the subject. The introduction is well-composed and has been developed on the right lines. The most appropriate methodology has been used for analysis of data and information. The results have been reported well. A logical sequence of interpretations has been followed and developed scientifically. The discussion has been well brought out and the cogency of arguments are well thought out. The discussion is comprehensive and complete. References are as required. May be accepted for publication .

Reviewer #2: The manuscript addresses a significant issue with a clear methodology and relevant findings. However, it requires several improvements in clarity, structure, and presentation to enhance its readability and impact. Please find the comments below:

Abstract

The abstract clearly outlines the study's objective, methodology, key findings, and implications. However, it could be more concise, focusing on the most critical points. Anyway I suggest that author must reduce redundancy, such as "The study applied a mixed-methods approach to investigate the effects of fuel price hikes on the livelihoods of coastal fisherfolk using the sustainable livelihoods framework" can be shortened. Also authors need to highlight the main findings more succinctly, possibly breaking them into bullet points for clarity.

Introduction

The introduction provides a thorough background on the issue, including relevant statistics and references. The context of global oil prices affecting local economies is well established. Also the Sustainable Livelihoods Approach (SLA) is well explained and relevant to the study. However, I think 1 and 2 should be combined as "Introduction". I think authors need to work really hard to restructure this two sections.

I also suggested that this section is dense with information. Please consider breaking it into smaller paragraphs for better readability. Also author need to clearly state the research questions earlier in the introduction to guide the reader. Please also add more recent studies to support the discussion on SLA and its application to fisheries.

M&M

I think authors need to provide more details on the sampling techniques and why 220 households and 20 stakeholders were chosen. Also author need to discuss any limitations of the methods used and how they were addressed.

Results

I request the authors please use more visual aids like graphs and tables to summarize key data points. Also authors must provide more direct quotes from interviews to support qualitative findings.

Discussion

Please ensure that each finding is discussed in relation to the research questions. I think authors need to highlight the practical implications of the findings for policymakers and stakeholders more explicitly. Additionally authors need to include specific recommendations for future research, and address the limitations of the study briefly and suggest how future studies could overcome these.

Frankly speaking, this work lacks of any interesting point and to really sure what is a really new findings from the work. Please emphasize that and explain more carefully.

6. PLOS authors have the option to publish the peer review history of their article (what does this mean?). If published, this will include your full peer review and any attached files.

Reviewer #1: No

Reviewer #2: No

---

## [Author Response · Author response to Decision Letter 0]

15 Sep 2024

Academic editor

Response

The manuscript has been formatted in relation to PLOS ONEs style requirements.

2.Note from Emily Chenette, Editor in Chief of PLOS ONE, and Iain Hrynaszkiewicz, Director of Open Research Solutions at PLOS: Did you know that depositing data in a repository is associated with up to a 25% citation advantage (https://doi.org/10.1371/journal.pone.0230416)? If you’ve not already done so, consider depositing your raw data in a repository to ensure your work is read, appreciated and cited by the largest possible audience. You’ll also earn an Accessible Data icon on your published paper if you deposit your data in any participating repository (https://plos.org/open-science/open-data/#accessible-data).

Response

Thank you for the information

3. You indicated that ethical approval was not necessary for your study. We understand that the framework for ethical oversight requirements for studies of this type may differ depending on the setting and we would appreciate some further clarification regarding your research. Could you please provide further details on why your study is exempt from the need for approval and confirmation from your institutional review board or research ethics committee (e.g., in the form of a letter or email correspondence) that ethics review was not necessary for this study? Please include a copy of the correspondence as an ""Other"" file.

Response 

Over the past few years l have established working relationships with fisherfolks in the Central Region of Ghana and has built trust with the communities and leadership of the fisherfolks for successful work. The already established relationship means easy entry into the communities and being affiliated with an academic community gives the team credibility to conduct ethically appropriate research with the rights and safety of the research participants and the researcher intact. Thus, the collaboration between the researchers and the case study communities in the Central Region of Ghana has developed into a long-standing respectful and reciprocal relationship. 

An introductory letter has also been attached File.

5. PLOS requires an ORCID iD for the corresponding author in Editorial Manager on papers submitted after December 6th, 2016. Please ensure that you have an ORCID iD and that it is validated in Editorial Manager. To do this, go to ‘Update my Information’ (in the upper left-hand corner of the main menu), and click on the Fetch/Validate link next to the ORCID field. This will take you to the ORCID site and allow you to create a new iD or authenticate a pre-existing iD in Editorial Manager. Please see the following video for instructions on linking an ORCID iD to your Editorial Manager account: https://www.youtube.com/watch?v=_xcclfuvtxQ.

RESPONSE

ORCID iD 0009-0004-4801-0985

0009-0004-4801-0985

6.Please amend your manuscript to include your abstract after the title page.

Thank you, this has been done

7.We note that Figure 4 in your submission contain [map/satellite] images which may be copyrighted. All PLOS content is published under the Creative Commons Attribution License (CC BY 4.0), which means that the manuscript, images, and Supporting Information files will be freely available online, and any third party is permitted to access, download, copy, distribute, and use these materials in any way, even commercially, with proper attribution. For these reasons, we cannot publish previously copyrighted maps or satellite images created using proprietary data, such as Google software (Google Maps, Street View, and Earth). For more information, see our copyright guidelines: http://journals.plos.org/plosone/s/licenses-and-copyright.

RESPONSE

Fig 4 has been removed.

RESPONSE

All the necessary changes have been effected. Thank you

Reviewer 2 Comments 

1.The abstract clearly outlines the study's objective, methodology, key findings, and implications. However, it could be more concise, focusing on the most critical points. Anyway I suggest that author must reduce redundancy, such as "The study applied a mixed-methods approach to investigate the effects of fuel price hikes on the livelihoods of coastal fisherfolk using the sustainable livelihoods framework" can be shortened. Also authors need to highlight the main findings more succinctly, possibly breaking them into bullet points for clarity

Response 

I thank the reviewer for pointing this out. The necessary improvements have been made.

2.The introduction provides a thorough background on the issue, including relevant statistics and references. The context of global oil prices affecting local economies is well established. Also the Sustainable Livelihoods Approach (SLA) is well explained and relevant to the study. However, I think 1 and 2 should be combined as "Introduction". I think authors need to work really hard to restructure this two sections.

I also suggested that this section is dense with information. Please consider breaking it into smaller paragraphs for better readability. Also author need to clearly state the research questions earlier in the introduction to guide the reader. Please also add more recent studies to support the discussion on SLA and its application to fisheries.

Response

Section 1 and 2 have been combined as introduction. The section has also been broken into smaller paragraphs to make it easy to read. All the other necessary corrections has been made.

3. I think authors need to provide more details on the sampling techniques and why 220 households and 20 stakeholders were chosen. Also author need to discuss any limitations of the methods used and how they were addressed.

Response

The sampling techniques has been further explained see page 13 -14.

The sample size used for the study is actually 320 not 220 as initially reported by the author. It’s a typological error, kindly pardon me. The process leading to the selection of the 320 participants is explained in pages 13-15.

4. I request the authors please use more visual aids like graphs and tables to summarize key data points. Also authors must provide more direct quotes from interviews to support qualitative findings.

Response 

Thank you for this comment, l have worked on the comments accordingly , a new chart has been added and some interview quotations have also been added.

5. Please ensure that each finding is discussed in relation to the research questions. I think authors need to highlight the practical implications of the findings for policymakers and stakeholders more explicitly. Additionally authors need to include specific recommendations for future research, and address the limitations of the study briefly and suggest how future studies could overcome these.

Response 

Thank you for this valuable comment. The discussion section has been worked on. Kindly see the discussion section.

6.Frankly speaking, this work lacks of any interesting point and to really sure what is a really new findings from the work. Please emphasize that and explain more carefully.

Response

This paper contributes to the literature on global fuel price hikes and its implications for marine fisheries sustainability by investigating the effects of oil price shocks on fisheries’ livelihoods in SSF communities in Ghana. The study considers the impacts and the various coping strategies fisherfolk deploy in response to fuel price shocks. The study contributes to the literature on coastal livelihoods, ocean governance, and natural resources management. Ghana is both an oil importer and exporter; hence, examining the effects of escalating global oil prices on the local fishing industry would interest the government, development practitioners, and policymakers. The results from the study could further help to devise a more specific stimulus package tailored to cater to the unique needs of the sector involved.

---

## [Decision Letter · Decision Letter 1]

19 Nov 2024

PONE-D-23-38320R1Effect of rising fuel prices on small-scale fisheries livelihoods and marine sustainability in Ghana.PLOS ONE

Dear Dr. Owusu,

Thank you for submitting your manuscript to PLOS ONE. After careful consideration, we feel that it has merit but does not fully meet PLOS ONE’s publication criteria as it currently stands. Therefore, we invite you to submit a revised version of the manuscript that addresses the points raised during the review process.

**ACADEMIC EDITOR:**Please see attached file. Kindly address comments raised in detail, and provide responses to the each comment in your reply (Please, this is compulsory)

We look forward to receiving your revised manuscript.

Kind regards,

Charles Odilichukwu R. Okpala, PhD

Academic Editor

PLOS ONE

Journal Requirements:

Additional Editor Comments:

Kindly attend to the comments raised.

Reviewers' comments:

Reviewer's Responses to Questions

**Comments to the Author**

1. If the authors have adequately addressed your comments raised in a previous round of review and you feel that this manuscript is now acceptable for publication, you may indicate that here to bypass the “Comments to the Author” section, enter your conflict of interest statement in the “Confidential to Editor” section, and submit your "Accept" recommendation.

Reviewer #3: (No Response)

2. Is the manuscript technically sound, and do the data support the conclusions?

Reviewer #3: Yes

3. Has the statistical analysis been performed appropriately and rigorously? 

Reviewer #3: Yes

4. Have the authors made all data underlying the findings in their manuscript fully available?

Reviewer #3: Yes

5. Is the manuscript presented in an intelligible fashion and written in standard English?

Reviewer #3: Yes

6. Review Comments to the Author

Reviewer #3: (No Response)

7. PLOS authors have the option to publish the peer review history of their article (what does this mean?). If published, this will include your full peer review and any attached files.

Reviewer #3: No

---

## [Author Response · Author response to Decision Letter 1]

18 Dec 2024

1.This should be part of the recommendation drawn from your study. It should not be included here as a specific objective

Response 

I thank the reviewer for this important comment, I have deleted objective 3 accordingly. 

2. Please briefly include the livelihood supported by the sector i.e. both direct and indirect/families beneficiary that relay on the aquatic resources.

Response 

I thank the reviewer for this observation. The fisheries sector directly support the livelihoods of men and women in the coastal communities. The men are engaged in fishing , the women are involved in processing and trading of fish. Indirectly the sector also support the livelihoods of boat builders, food vendors etc Please page 9, line 226-230.

3. Why the interview was conducted with stakeholders only? It should have been conducted with selected fishers too, to elaborate the quantitative finding.

Response

The stakeholders includes fisherfolk, government officials and NGOs. This is further explained – please see page 11, line 287-289.

4. Please compare the change in fuel price/a single production of the fishers before and after the fuel price change in GHANA

Response

Thank you, this has been done.

5.If you had employed quantitative survey, why you did not measure the livelihoods of the fishers using indicators for these different capital assets.

Response

I thank the reviewer for this observation. Unfortunately, l did not measure the livelihood of the fishers in relation to the capital assets due to the limited scope of study. Since the study is limited in its scope l decided to focused on describing how the fuel price hikes have negatively affected the livelihood assets of fisherfolk.

6. If you have a data, please include one table that indicate fuel consumption of fishers/trip, boat horsepower with their respective fuel consumption, and enquiring cost per production time and production in tone/kg. The profitability to support their livelihood should be related with their income and expense

Response

Different boats catch different species using boat engine capacity of 40hp. In addition, the number of fishing days differ from boat to boat. For instance the quantity of fish for a set net gear will differ from that beach seine because they all target different species and the weight differ in terms of the fish. So the average fish catch for 40hp per trip will be difficult to provide.

I have however included information provided by one of the interviewed canoe owner in relation to fuel consumption per fishing trip with 40 hp / cost of production and volume of fish landed in kg. Please see page 17, line 384 -393.

7. Please include some points on sustainability either here or in recommendation section

Response 

This have been done, please see page page 25 , line 617.

8. Please remove the reference from this section. As this is your conclusion and recommendation from the study.

Response 

Thank you, this has been done.

---

## [Editor Report · Decision Letter 2]

26 Dec 2024

Effect of rising fuel prices on small-scale fisheries livelihoods and marine sustainability in Ghana.

PONE-D-23-38320R2

Dear Dr. Owusu,

We’re pleased to inform you that your manuscript has been judged scientifically suitable for publication and will be formally accepted for publication once it meets all outstanding technical requirements.

Kind regards,

Charles Odilichukwu R. Okpala, PhD

Academic Editor

PLOS ONE
---

## [Editor Report · Acceptance letter]

2 Jan 2025

PONE-D-23-38320R2 

PLOS ONE

Dear Dr. Owusu, 

I'm pleased to inform you that your manuscript has been deemed suitable for publication in PLOS ONE. Congratulations! Your manuscript is now being handed over to our production team.

Kind regards, 

on behalf of

Dr. Charles Odilichukwu R. Okpala 

Academic Editor

PLOS ONE